# Self-Supervised Similarity Learning for Digital Pathology

Jacob Gildenblat[1,2] and Eldad Klaiman[3✉]

[1] SagivTech Ltd.
[2] DeePathology.ai
[3] Pathology and Tissue Analytics, Pharma Research and Early Development, Roche
Innovation Center Munich
eldad.klaiman@roche.com

**Abstract.** Using features extracted from networks pretrained on ImageNet is a common practice in applications of deep learning for digital pathology. However it presents the downside of missing domain specific image information. In digital pathology, supervised training data is expensive and difficult to collect. We propose a self-supervised method for feature extraction by similarity learning on whole slide images (WSI) that is simple to implement and allows creation of robust and compact image descriptors. We train a siamese network, exploiting image spatial continuity and assuming spatially adjacent tiles in the image are more similar to each other than distant tiles. Our network outputs feature vectors of length 128, which allows dramatically lower memory storage and faster processing than networks pretrained on ImageNet. We apply the method on digital pathology WSIs from the Camelyon16 train set and assess and compare our method by measuring image retrieval of tumor tiles and descriptor pair distance ratio for distant/near tiles in the Camelyon16 test set. We show that our method yields better retrieval task results than existing ImageNet based and generic self-supervised feature extraction methods. To the best of our knowledge, this is also the first published method for self-supervised learning tailored for digital pathology.

**Keywords:** Deep Learning · Self-Supervised · Similarity Learning · Digital Pathology.

## 1   Introduction

There are many applications of Deep Learning for digital pathology [5]. Some examples of such applications are tissue segmentation [8], whole slide images (WSI) disease localization and classification [9, 2], cell detection [13] and virtual staining [7].

WSI are typically large, and in full resolution may typically contain 1 billion pixels or more. It is therefore common practice to divide the WSI into tiles and analyse the individual tiles in order to sidestep the memory bottleneck. Often convolutional neural networks (CNN) pretrained on ImageNet are used to

extract features from these tiles. For example in [2] features are extracted using the ResNet-50 [4] network trained on ImageNet, and then a semi supervised classification network is trained on these features using multiple instance learning (MIL). This is done because these features can function as rich image descriptors. In some cases, training networks from scratch for histopathological images is not feasible due to the weak nature of the labeling or the limited amount of annotated data.

The use of networks pretrained on ImageNet is common mostly because of the availability of these pretrained networks. Imagenet pretrained networks are typically trained with supervised learning on a large and variable annotated dataset of natural images with 1000 categories. There is a lack of similar available annotated datasets that capture the natural and practical variability in histopathology images. For example, even existing large datasets like Camelyon consist of only one type of staining (Hematoxylin and Eosin), one type of cancer (Breast Cancer) and only two classes (Tumor/Non-Tumor). Histopathology image texture and object shapes may vary highly in images from different cancer types, different tissue staining types and different tissue types. Additionally, histopathology images contain many different texture and object types with different domain specific meanings (e.g stroma, tumor infiltrating lymphocytes, blood vessels, fat, healthy tissue, necrosis, etc.).

In many domains, the shortage in annotated datasets has been addressed by unsupervised and self-supervised methods [6], which have been shown to hold potential for training networks that can serve as useful feature extractors. Self-supervised learning is a subset of unsupervised learning methods in which CNNs are explicitly trained with automatically generated labels [6].

In [14] the authors propose learning to colorize images, and then use the learned network as a feature extractor. In [3] the authors propose to rotate images and then learn to predict the rotation angle that serves as a synthetic label to train the network. A recent method that obtains state of the art results on standard (non-pathology) feature extraction benchmarks is described in [12]. The described method tries to discriminate between images in the dataset by predicting the index of the input image.

Generation of medical annotated data such as pathological images is especially time consuming and expensive, therefore we would like to use self-supervised approaches whenever possible. In the case of digital pathology, an intrinsic property of the tissue image is its continuity. This means that two tissue areas that are near each other are more likely to be similar than two distant tiles. A straightforward way for self-supervised image labeling for digital pathology images would then be labeling pairs of similar and non-similar images based on their spatial proximity. This would enable generating automatically labeled data for very large and diverse datasets.

Datasets with similarity labels can be used to train networks using a method called similarity learning in which an algorithm is trained by examples to measure how similar two objects are. Some important applications of these types of methods include image search and retrieval, visual tracking, and face verifica-

tion. A common family of network architectures for similarity learning is siamese networks [1]. They consist of two or more identical sub networks sharing weights and trained on pairs or larger sets of images in order to rank semantic similarity and distinguish between similar and dissimilar inputs. Training siamese networks requires a way to generate pairs of images with a label indicating if they are similar or not.

We note that there is a noticeable lack of self-supervised methods that exploit domain specific characteristics existing in gigapixel histopathology WSIs. A common method to generate image descriptors given the shortage of labeled datasets is the use of pretrained ImageNet networks. However these descriptors lack domain specific information from the target dataset e.g. digital pathology. We propose a novel self-supervised learning method which leverages the intrinsic spatial continuity of histopathological tissue images to train a model that generates domain specific image descriptors. We apply our method on the Camelyon16 dataset in order to perform an image retrieval task for tumor areas. And we validate our approach by comparing descriptor distance of similar and dissimilar expert labeled images. We compare our results to a state of the art self-supervised method.

## 2   Proposed Approach

We observe that WSIs have an inherent spatial continuity. Spatially adjacent tiles are typically more similar to each other than distant tiles in the image. In order to generate our training dataset we define a maximum distance between pairs of tiles to be labeled as similar, and a minimum distance between tiles to be labeled as non-similar based on the intrinsic sizes of histo-pathological regions in the dataset. For each tile in the dataset other similar or non-similar tiles are sampled based on the predefined distance thresholds creating a dataset of automatically labeled pairs. This sampling strategy allows easily creating a very large and diverse set of pairs sampled from histopathology WSIs without the need for any manual annotation.

Using this dataset we train a siamese network for image similarity leveraging this spatial continuity. The network used consists of two identical sub networks sharing weights and trained on pairs of similar and dissimilar inputs. As a training loss for the siamese network we use a contrastive loss on pairs of images 1.

$$L_{contrastive} = (1 - y)L_2(f_1 - f_2) + y \times \max(0, m - L_2(f_1 - f_2)), \qquad (1)$$

where $f_1$, $f_2$ are the outputs of two identical sub networks. $y$ is the ground truth label for the image pair: 0 if they are similar, 1 if they are not similar.

The Camelyon16 test dataset includes manual expert annotations for tumor areas. In order to evaluate the performance of the network in capturing the histopathological features in the image descriptors, we use these ground truth

annotations to form pairs of similar and non-similar tiles by pairing tumor labeled tiles with tumor and non-tumor tiles respectively. We calculate the L2 distance between image descriptors for each pair in the test dataset. We define the global Average Descriptor Distance Ratio (ADDR) as the ratio of the average descriptor distance of non-similar pairs and the average descriptor distance of similar pairs for all pairs in the test dataset.

As an additional evaluation metric we measure the ability of the learned network to perform a pathology image retrieval task on the Camelyon16 test set. Every tile extracted from the Camelyon16 testing set is marked as "tumor" if it lies entirely inside the expert tumor annotation area. A nearest neighbor search on feature vectors is performed on each tile, constraining the search to tiles from other slides in order to more robustly assess descriptor generalization across different images. We report the percentage of correct nearest neighbor tumor tile retrieval.

We compare our self-supervised method generated image descriptors to a ResNet-50 pretrained on ImageNet as well as to a state of the art self-supervised learning method called Non-Parametric Instance Discrimination (NPID) [12]. NPID tries to discriminate between all the images in the datasets by using the index of the input image as a synthetic label to train the network.

## 3    Experiments

In this section we describe the datasets and experiments and give more detailed information about the implementations and the results.

### 3.1    Datasets

All experiments were done on tiles extracted from the Camelyon16 training dataset at x10 resolution. The Camelyon16 training dataset contains 270 breast lymph node Hematoxylin and Eosin (H&E) stained tissue WSIs. We validate and assess our method on the Camelyon16 testing set which contains 130 H&E stained WSIs.

Our training and testing datasets were created by extracting non overlapping tiles of size 224x224. We defined a maximum distance of 1792 pixels ( 2mm) between two tiles for them to be considered similar, and a minimum distance of 9408 pixels ( 10mm) between two tiles for them to be considered non-similar. We choose this threshold based on the histopathological definition of a macrometastasis which is typically larger than 2mm [11]. Sampling 32 pairs of near tiles and 32 pairs of distant tiles per tile in the dataset yielded 70 million pairs of which half are labeled similar, the others are labeled non-similar. A sample of similar and non-similar pairs form the training dataset can be seen in Fig. 1

For the testing of our method we generated a dataset from the Camelyon16 test dataset, by sampling 8 near tiles and 8 distant tiles per tile using the expert ground truth as described in the proposed approach section. This resulted in 1,385,288 pairs of similar tiles and 1,385,288 non-similar tiles.

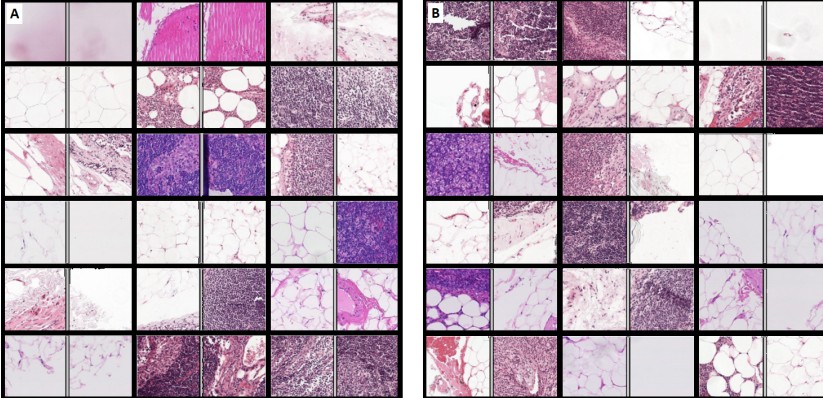

**Fig. 1.** Visualization of sampled tile pairs. (A) - Pairs of tiles close to each other, labeled as similar images. (B) - Pairs of tiles far from each other, labeled as unsimilar images.

## 3.2  Implementation Details

We trained a siamese network consisting of two branches of modified ResNet-50 [4] with the last layer (that normally outputs 1,000 features) is replaced with a fully connected layer of size 128. For training our siamese network we use the Adam optimizer with the default parameters in PyTorch (learning rate of 0.001 and betas of 0.9,0.999), and a batch size of 256. For data augmentation, we used random horizontal and vertical flips, a random rotation up to 20 degrees, and a color jitter augmentation with a value of 0.075 for brightness, saturation and hue. The network was trained for 24 hours using 8 V100 GPUs on the Roche Penzberg High Performace Compute Cluster using a PyTorch DataParallel implementation.

In experiments using the ImageNet pretrained ResNet-50 we extract features from the second last layer, with a length of 2048.

Training of the NPID network was also performed using 8 V100 GPUs for 24 hours and loss convergence was observed.

In the application of the networks on the test set, we normalize the image by matching the standard deviation and the mean of each channel in the LAB color space of the source tile with a preselected target tile. This provides a sort of simple stain normalization for the WSIs.

## 3.3  Results and Comparison

In the experiment measuring L2 distance between descriptors of distant and near tiles we report the global ADDR for ImageNet pretrained ResNet-50, the NPID method and our proposed approach. Results can be seen in Table 1.

**Table 1.** L2 distance ratio between descriptors of distant and near tiles.

| Method | Global ADDR |
|---|---|
| ResNet-50 pretrained on ImageNet | 1.38 |
| Non-Parametric Instance Discrimination | 1.28 |
| Ours | **1.5** |

The result of this experiment indicate that our method outperforms the benchmark methods in the task of separating similar and non-similar tiles in the descriptor space.

In the tumor tile retrieval experiment, 3809 tumor tiles were extracted from the test dataset based on expert ground truth annotation as described in the proposed approach section. The target class tumor comprises only 3% of the entirety of the tiles searched in the test set. We report the percentage of correctly retrieved tiles in Table 2.

**Table 2.** Results for tumor tile retrieval.

| Method | Ratio of retrieved tumor tiles |
|---|---|
| ResNet-50 pretrained on ImageNet | 26% |
| Non-Parametric Instance Discrimination | 21% |
| Ours | **34%** |

It can be seen from the results of the retrieval task that our method substantially outperforms both the ImageNet pretrained network as well as the NPID method. Additionally, examples of nearest neighbor retrievals from our network in this experiment can be seen in Fig. 2.

## 4    Conclusion and Discussion

We present a novel self-supervised approach for training CNNs for the purpose of generating visually meaningful image descriptors. In particular, we show that using this method for images in the digital pathology domain yields substantially better image retrieval results than other methods on the Camelyon16 dataset. We evaluate and compare the performance of our method with other benchmark methods in a retrieval task and a descriptor distance metric on the Camelyon16 test set. Our method presents potential to create better feature extraction algorithms for digital pathology datasets where labels for a supervised training are hard to obtain. We believe that this work can be a first step towards the adaptation of self-supervised methods for image descriptor generation in digital pathology instead of using features from networks pretrained on ImageNet.

A disadvantage of the spatial similarity sampling strategy is that in some cases pairs of images are not accurately labeled. For example in transitions between different functional histological areas in the image there are by definition

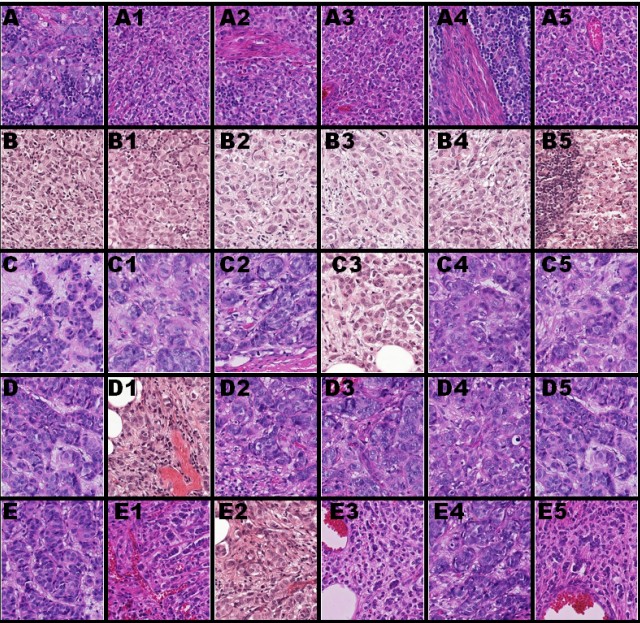

**Fig. 2.** Examples results for 5 tumor query tiles (A, B, C, D, E) in the image retrieval task and the 5 closest retrieved tiles from slides other than the query slide (A1-A5, B1-B5, C1-C5, D1-D5, E1-E5), ranked by distance from low to high, using tile descriptors generated by our method. Its interesting to note that even though some retrieved tiles look very different than the query tile (e.g. C3 and C) all of the retrieved tiles except A4 have been verified by an expert pathologist to contain tumor cells (i.e. correct class retrieval).

spatially proximal tiles that are visually different. In other cases two distant tiles can be visually similar because they are part of distant areas of the same histopathological function. This effect creates an inherent labeling noise in the dataset. A reasonable assumption in many WSIs is that region borders typically occupy less area in the image than the regions themselves so the label noise is substantially lower than the correct labels. Additionally, due to the statistical characteristics of their training process, deep learning methods have been shown to be predominantly robust to label noise even in extreme cases [10].

Future work will include verification strategies for sampled pairs, and new sampling strategies for self-supervised similarity learning as well as hyper-parameter tuning and exploration of the proximity thresholds in the dataset generation process.

## Acknowledgements

The authors would like to thank Amal Lahiani for her invaluable insights and constructive review.

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
