# OpenReview forum: "Self-Supervised Similarity Learning for Digital Pathology"
_MICCAI.org/2019/Workshop/COMPAY — COMPAY 2019_

### Official Review · AnonReviewer1 · 2019-07-25
**self-supervised image descriptor generation in digital pathology**

**Rating:** 8
**Confidence:** 3

**Review:**

The paper presents self-supervised method for feature extraction by similarity learning using siamese network. The presented approach is interesting and has been tested on a publicly available Chamelyon16 data set. The authors utilize distance of image patches as a way to measure similarity and test their approach using average descriptor distance ratio and image retrieval. Although the paper is well-written but there are a few parts which need further explanation for example, it would be good to show the network architecture and the sub-networks sharing weights which are trained on pairs of similar and non-similar inputs. Secondly, the authors show image retrieval on tumor samples, can the authors comment on how image retrieval will work on non-tumor regions.

---

### Official Review · AnonReviewer4 · 2019-08-15

**Rating:** 6
**Confidence:** 3

**Review:**

This paper presents an interesting self-supervised method to generate image descriptor features in digital pathology. The method leverages the intrinsic spatial continuity of WSIs to select similar and non-similar image patches based on their spatial distance from each other in a WSI. The method is applied to the Camelyon16 dataset and compared to a similar self-supervised method and a method pre-trained on ImageNet.

The paper is well written, however I do have some concerns, questions and suggestions:
- Would this method also work when trained with a smaller batch size? A batch of 256 224x244 pixel patches requires considerable GPU memory.
- A visualization of the Siamese network would clarify the paper.
- Please describe more clearly how your method was applied to the Camelyon16 test set. As I understand the method is trained using similarity/non-similarity based on distance of patches in a WSI. However, when applied to the test set you find the most similar patches (from a different WSI) to a specific tumor-labeled patch. Regarding matching the non-tumor patches I assume that you also find similarity results here based on stroma, fat, blood vessels, etc.
- The results only show the percentage of retrieved tumor tiles (34%). What was the amount of false positives here?-
How do pairs of similar non-tumor tissue look?

---

### Official Review · AnonReviewer3 · 2019-08-19
**Self-Supervised Similarity Learning for Digital Pathology**

**Rating:** 5
**Confidence:** 5

**Review:**

Two contributions of the paper are:
1) Applying Siamese Networks: Use of siamese networks to find similar and dissimilar patches.  As the authors also mention, it is a known technique for similarity matching in computer vision applications. This paper contribution is the application (first time?, I am not familiar with other papers) for histopathology use case.

2) Training sampling strategy:  To facilitate self-supervised similarity learning,  a training data sampling strategy making use of histopathology domain specific biologically meaningful spatial continuity constraint in gigapixel WSIs. It is an interesting and good idea. Along these lines, the authors have created a similar/dissimilar ground truth dataset for Camelyon-16 dataset.

I believe the interesting and main contribution in this paper  is  the sampling strategy and the application of siamese networks to DP domain is secondary (obvious and incremental).

In reference to the experiments and validation results, the comparative tables are with two metrics, for three different networks:(pretained ResNet50 siamese network on ImageNet data b) NPID trained on the created  histopathology dataset  and c) Modified Restnet-50 siamese network with histopathology dataset).

The improvements (table 1 and 2) can be due to any of these two reasons 1) retraining the siamese network with histopathology dataset 2)  Proposed sampling strategy.

If I understand it right, think the experiments are not appropriate to evaluate and demonstrate the improvements from the proposed sampling strategy.  The comparison to pretrained ImageNet or NPID method are marginal and irrelevant to evaluate the primary hypothesis. As the authors also mention in the discussion section, the training data constructed with such a sampling strategy can be quite noisy (due to tumor spatial heterogeneity and tissue variability).

 I think a good experiment to evaluate the proposed sampling strategy will be:
1. To create a training set and test from the Cameylon-16 ground truth (similarity pair) dataset (1,385,288 pairs of similar tiles and 1,385,288 non-similar tiles mentioned in the 3.1 last paragraph).
2. Using the training set, retrain the siamese network and test on the test dataset. And show the result similar and dissimilar images.
3. Change the training set with the proposed sampling strategy (70 million pairs of which half are labeled similar, the others are labeled non-similar). Retrain the siamese network. Using the siamese network, for the same test set used above, extract the similar and dissimilar images.
4. Compare the results from A) and B).

If the authors can update with the results for the above experiment, can be considered for acceptance.

---

### Official Review · AnonReviewer2 · 2019-08-22
**Review Self-Supervised Similarity Learning for Digital Pathology**

**Rating:** 6
**Confidence:** 5

**Review:**

This paper proposes to apply self-supervised learning principles and siamese networks to build compact feature vectors for image retrieval in digital pathology.
The main idea is to exploit spatial continuity in whole-slide images to generate large training sets of similar/dissimilar patches.

I think the idea is interesting and deserves to be presented during this workshop. However, I do have some questions and suggestions:
* Last paragraph of Introduction (Page 3) is a repetition of what is said previously, I think the Introduction could be reorganized.
* The impact of the maximum/minimum distance (Section 2, page 3) and hyper-parameters (Section 3.2) are not studied, so current values look a bit arbitrary.
* Justify ADDR (Section 2) with respect to other metrics and practical use (and move this to Section Experiments).
* For the baseline, explain why are features extract from the second last layer (Section 3.2)
* NPID description (page 4): the "index of the input image" is not well described.
* Table 1 page 6: It would be nice to include results with different maximum/minimum distance values, tile overlapping for similar pairs, and maybe also random pairs to train siamese networks as a baseline.
* Table 2 page 6:  Please clarify the protocol. Is it top-K (K=5 as in Figure 2) retrieval accuracy ?

---

### Decision · Program_Chairs · 2019-08-20

Accept